# Analysis of Fibre Cross-Coupling Mechanisms in Fibre-Optical Force Sensors [note 1]

**DOI:** 10.3390/s21072402

**Published:** 2021-03-31

**Authors:** Christian-Alexander Bunge, Jan Kallweit, Levent Colakoglu, Thomas Gries

**Affiliations:** 1Faculty for Digital Transformation (F-DIT), HTWK Leipzig University of Applied Sciences, Zschochersche Str. 69, D-042289 Leipzig, Germany; 2Institute for Textile Technology (ITA), RWTH Aachen University, Otto-Blumenthal-Str. 1, 52074 Aachen, Germany; jan.kallweit@ita.rwth-aachen.de (J.K.); levent.colakoglu@rwth-aachen.de (L.C.); thomas.gries@ita.rwth-aachen.de (T.G.)

**Keywords:** pressure sensor, fibre sensor, cross coupling, ray tracing, sensor array

## Abstract

The force-enhanced light coupling between two optical fibres is investigated for the application in a pressure or force sensor, which can be arranged into arrays and integrated into textile surfaces. The optical coupling mechanisms such as the influence of the applied force, the losses at the coupling point and the angular alignment of the two fibres are studied experimentally and numerically. The results reveal that most of the losses occur at the deformation of the pump fibre. Only a small percentage of the cross-coupled light from the pump fibre is actually captured by the probe fibre. Thus, the coupling and therefore the sensor signal can be strongly increased by a proper crossing angle between the fibres, which lead to a coupling efficiency of 3%, a sensitivity improvement of more than 20 dB compared to the orthogonal alignment of the two fibres.

## 1. Introduction

Force and pressure sensors are needed in a variety of different fields, for example, in industrial or medical devices [1] or in seats and furniture [2,3]. Many of these sensors rely on electrical effects such as capacitance (e.g., [4,5,6,7]) or piezo-resistance (e.g., [8,9,10,11]). These methods are robust but can only be used in certain applications and environments. They can be sensitive to electromagnetic interference and problematic in hazardous areas, under humid or wet conditions [12]. Optical sensors and especially optical fibre sensors can overcome these problems and offer more benefits such as simple integration into textiles and distributed sensing [13,14,15]. One approach is to evaluate the optical coupling between two fibres as an indicator for the applied force. In [16], micro fibres are used due to their increased evanescent fields, which enhance the coupling. The use of single-mode fibres and even fibre bundles may lead to still higher coupling [17]. Most of these sensing schemes rely on glass optical fibres, which show low attenuation and small diameters, but are neither bio-compatible nor robust enough for textile integration [18].

There have been several proposals for optical force sensors using polymer optical fibres (POF), which are more robust. Initially, POF pressure sensor technology was—and still is—used in the field of robotics [19,20,21,22]. In [23] for instance, the losses induced by macro bending in a POF arrays have been used as sensor signal. In [24], the authors proposed a similar concept of a fibre-based textile touchpad. But this sensor array offered spatially resolved two-dimensional force sensitivity with multi-touch capability. In contrast to [23], not the losses were evaluated, but the coupling between two crossing fibres. Similar concepts had already been proposed in [19,20] and also by Schoenwald et al. [21], which relied on two different layers of crossing fibres. Rothmaier et al. presented a textile integration of this pressure sensor [25], but none of the aforementioned methods were fully compatible with textile technology nor did they provide simultaneous sensing at different locations. The use of soft or elastomeric materials for POF sensing in general [26,27,28] and for this type of pressure sensing via POF [24] has shown promise and is therefore also addressed in the paper. Apart from these aspects, the authors investigated in [24] the coupling effect between the crossing fibres in more detail, studying the influence of different materials and mechanical properties of the fibre. These proof-of-concept experiments showed the applicability of the effect, but did not investigate the cross-coupling mechanism much further. In [29], the authors studied the coupling effects and their influences experimentally in order to increase the sensitivity.

In this contribution, the cross coupling is further investigated experimentally and modelled numerically for a better understanding of the coupling mechanism itself. The results indicate that the sensitivity can be much increased by a combination of deformation and mode coupling at the fibre crossing as well as different alignments and crossing angles between the fibres.

## 2. Materials and Methods

In this section, the optical cross-coupling effect between two fibres is analysed and explained in detail. From these assumptions, expected trends and influencing parameters are deduced, which will be systematically investigated by experiments. Therefore test fibres have to be produced and the measurement setups and evaluation methods will be implemented. Finally as way to further understand the coupling process, a numerical model is set up that simulated the cross-coupling effect considering the two most important optical effects, the optical leakage from the pump fibre and the mode conversion within the probe fibre.

### 2.1. Optical Fibre Cross Coupling

The fundamental idea of the pressure sensor is illustrated in Figure 1a,b. It consists of a pump fibre that is illuminated by a light source and crossed with a probe fibre in order to receive a fraction of the light of the pump fibre by cross-coupling. This light is detected by a photodiode at the end of the probe fibre. With increasing pressure on the crossing point more light will be coupled into the probe fibre. Since only a small part of the pump light is cross-coupled into the probe fibre, there is still enough optical power for further sensitive crossing points behind the first crossing. Thus, multiple sensing points along the same fibre and with more than one pump fibre and even 2D sensor arrays are possible that can detect several pressure points simultaneously.

The actual optical coupling mechanism is based on two different effects: With applied pressure, the two fibres will deform and their surfaces will touch each other. The exact process of deformation of both fibres can be described by the Hertzian contact theory [30]. Therefore, light can exit the pump fibre through the interface, where both fibres touch each other.

But, according to reciprocity, all light that enters from the side into an ideal fibre will pass the fibre and exit further on. Therefore, scattering or another effect (e.g., luminescent particles as in solar concentrators [31] or plastic scintillating fibres [32]) that leads to a mode conversion must occur in the probe fibre. This can be deformation, which leads to scattering at the core interface, but also scattering within the core material is even more efficient to capture the light. For smaller crossing angles the change of propagation direction can be smaller, which enhances the coupling effect.

While the formation of the Hertzian contact is mainly influenced by the mechanical properties of the fibres and the supporting structure, the optical mode conversion in the probe fibre can be influenced by the optical setup.

### 2.2. Method

The coupling effect between the pump and probe fibre is studied in a systematic manner in order to optimise the sensitivity of the sensor. Therefore, soft optical fibres made from polyurethane (PUR) resin have been fabricated and several investigations been performed. The actual influence of the applied force on the cross-coupled optical power is studied first as a reference. Then, the losses are studied which also increase with applied force and show that only a small amount of the light is actually recollected by the probe fibre. Thus, the influence of the crossing angle is studied to optimise the reception.

#### 2.2.1. Fibre Preparation

Deformable polymer optical fibres made from polyurethane were produced according to the following procedure: Since only very short fibres were needed, a simple production process based on casting was chosen, which is known for a good surface quality [33]. The PUR resin “PUR-Gießharz 1770/330” from Modulor GmbH, Germany, was thoroughly filled into silicone hoses. These came from Deutsch & Neumann, Germany, and showed an inner diameter of D=2mm. The full hoses were left to rest at room temperature for at least 4 hours to allow the PUR core to harden. Then, the PUR core could be separated from the hose by cutting the silicone hose with a razor blade and peeling it off. The resulting fibre filaments showed a hardness of about 55 Sh-A, a refractive index of n=1.34 and a fibre diameter of 2 mm. Measurements of the optical attenuation revealed $\alpha \left(633\;\mathrm{nm}\right)=0.3637\;\frac{\mathrm{dB}}{\mathrm{cm}}$.

#### 2.2.2. Measurement Setup

In order to cross couple light between the two PUR fibres under a specific load, a coupling piece was 3D printed, in which the angle between the pump and probe fibre could be controlled in $15^{\circ }$ steps between $\alpha =15^{\circ }$ and $\alpha =90^{\circ }$. Figure 2 shows the coupling piece, the alignment of the two fibres and how the normal force was applied to the crossing point.

The probe fibre was guided through the lower bore, whereas the pump fibre was guided through one of the slightly higher boreholes. To press the fibres against each other, the coupling piece had a large hole from the top through which a pressure pin was guided to the cross section of the fibres (c.f. Figure 2c). The pressure pin could be loaded with a certain number of nuts, which increased the normal force applied onto the fibres by 0.28 N each.

The principle setup of the actual measurements is depicted in Figure 3. The LED “IF-E91D” launched the light with a power of $P_{in}\approx 201.5$ μW into the pump fibre. The actual coupling occurred after $L_{in}$ = 3 cm. Here, the originally coupled power has been decreased to 1563 μW due to the losses of 3 cm fibre ($a_{in}$ = 1.09 dB). Some part of the power is coupled into the probe fibre, where the signal $P_{probe}$ was detected by the photodiode IF-D91 (Industrial fibre Optics Inc, Temple, AZ, USA) at the fibre end. The LED was controlled by an Arduino Uno. A transimpedance amplifier (TIA) consisted of a LM358P operational amplifier (Texas Instruments Inc, Dallas, TX, USA) and was responsible for converting and amplifying the current of the photodiode into a measurable voltage, which could be evaluated as a power $P_{probe}$. Since the measurement was at the fibre end, the signal had suffered the additional fibre losses of $a_{probe}=1.09\;$dB for $L_{probe}=3\;$cm. The remaining power within the pump fibre was measured in the same way, but here the fibre length was considerably longer with $L_{pump}=16\;$cm, which resulted in additional losses of $a_{pump}=5.82\;$dB. This basic setup was used for all subsequent experiments with small adjustments regarding the coupling angle and amplification.

#### 2.2.3. Ray-Tracing Simulation

In order to better understand the actual coupling effect, a simple simulation model was developed that assumed a deformation of the two fibres so that an interface between them facilitated the entering of light from the pump to the probe fibre. This deformation was roughly modelled as a flat section of the otherwise circular cross section at a depth *d* of both fibres, which increased for larger applied forces according to the elasticity of the fibre material. Since the simulations are only used to explain the coupling mechanism and trend without the intention to predict precise values only the depth was varied without assuming specific forces. But it can be roughly estimated that the maximum applied force of 2 N resulted in a deformation in the range of d=30%. Both fibres were assumed to react similarly as both showed the same diameter and consisted of the same material. Figure 4 visualises the assumed deformation of both fibres.

The optical effect was modelled by a ray-tracing approach. At the start of the pump fibre, a set of $5\times 10^{5}$ rays was randomly chosen according to the location on the fibre end face (uniform near field) and the propagation directions (far field) obeyed a von-Mises distribution with μ=0 and κ=15.43 [34]. This approximated a so-called equilibrium-mode distribution (EMD) with a full width at half maximum of $20^{\circ }$ within the fibre. The rays were traced until they approached the location of the coupling at $L_{in}$. There, the rays entered the flattened fibre according to the assumed deformation. All rays that reached the flat side section of the fibre were recorded for further simulation. The remaining rays were traced until they reached the end of the coupling zone, where the remaining power $P_{pump}$ was evaluated.

The recorded positions and propagation directions of the rays, which had approached the flat coupling surface between both fibres, were used as starting points for a ray-tracing simulation in the probe fibre. In order to consider a crossing angle α between the fibres, the propagation directions of the rays were transformed by a rotation at the *x*-axis by α. Here, the rays started at the flat couple area and propagated through the fibre. The length and area of that coupling section depends on the amount of deformation and the crossing angle. Using geometrical relations, the width of the flat section *w* can be expressed as:(1)$$w=\frac{D}{2}\sqrt{d(2-d)}.$$

The area of the coupling section $A_{c}$ is then
(2)$$A_{c}=\frac{w^{2}}{\tan\alpha }$$
and the length $L_{c}$
(3)$$L_{c}=\frac{w}{\tan\alpha }.$$

Since many rays would not have been guided by the probe fibre—in particular for large tilting angles α—a certain amount of scattering at the interface was assumed. The scattering was modelled by a random, zero-mean normal-distributed change of the propagation direction Δθ at the interface between the two crossing fibres with the standard deviation $\sigma_{\Delta \theta }$, similarly to the method described in [35], where the probability density function p(Δθ) is defined by:(4)$$p\left(\Delta \theta \right)=\frac{1}{\sqrt{2\pi \sigma_{\Delta \theta }^{2}}}\exp\left(-\frac{\Delta \theta^{2}}{2\sigma_{\Delta \theta }^{2}}\right).$$

The rays that reached the end of the crossing region were evaluated in terms of $P_{probe}$.

These simulations were performed for different relative deformations *d*, crossing angles α and several scattering intensities in order to assess their respective influence. All simulations have been repeated 16 times. The average and standard deviation have been calculated for all results.

## 3. Results

First, the coupled power was measured for different loads as the main sensing effect. Then, further investigations followed for a better understanding, which included the actual evaluation of the coupling efficiency, but also of the remaining power in order to obtain insight on the overall efficiency of the setup. Then, the crossing angle was investigated more systematically since it provided huge room for improvement. Finally, these measurement results were compared to ray-tracing simulations, which modelled the optical coupling for different crossing angles, but also for different deformations and scattering intensities.

### 3.1. Influence of Applied Force

The measured influence of a normal force at the coupling point on the coupled power into the probe fibre is shown in Figure 5 for different crossing angles. The detected signal increases proportionally to the applied force starting from a minimum load of 0.28 N up to about 2 N. The fact that the linear increase does not start from zero can be explained by friction within the 3D-printed force-application rod. The results are consistent with the findings in [24] where melt-spun fibres out of harder materials like polymethyl methacrylate (PMMA) were used. It is obvious that the sensitivity of the signal increases for smaller coupling angles α.

The increase of the coupled power with increased force can be explained by the increasing contact area between the fibres when fibres are deformed elastically by the applied load. Above a certain force, the same applied force no longer results in the same increase in contact area due to viscoelastic material properties and the Hertzian contact theory of two cylindrical elements being pressed together.

If the losses within the pump fibre and the probe fibre are considered the coupling efficiency at the fibre crossing can be estimated. This The results are also shown in Figure 5. The shape of the curves are identical. Thus, a second ordinate on the right-hand side indicates the coupling efficiency directly at the crossing point. The coupling efficiency increases with applied force and exceeds 1.5% for shallow coupling angles and an applied force of about 2 N.

### 3.2. Correlation between Light Intensity in Illuminated and Probe Fibre

In a second investigation, the losses at the coupling points were studied by comparing the coupled and the remaining light intensities in both fibres. For this investigation, both fibres were coupled to a photodiode in order to measure the optical power at the fibre ends.

In Figure 6, the fraction of the remaining power in the pump fibre behind the crossing is plotted. This graph shows the amount of power that exits at the crossing point and reduces the remaining power. It is obvious that more light exits at the crossing point with applied force. But in contrast to the study of the coupled power, the coupling angle has hardly any influence. This indicates that the applied force leads to a deformation of the fibres, which facilitates a light leakage at the contact area. But the coupling angles show their main influence in capturing the leaked pump power into the probe fibre.

The shape of the plot also indicates that most of the effective deformation occurs at small applied forces below 0.5 N. For forces larger than 2 N, less than 20% of the inital power is only left. A slight saturation can be observed.

### 3.3. Total Efficiency

The findings above indicate that quite some power is lost during the coupling process. Whereas the coupling efficiency increases with more force and smaller coupling angles, it remains well below 10%. The remaining power within the pump fibre, on the other hand, reduces considerably with the applied force. Therefore, the total efficiency, has been plotted in Figure 7 in terms of losses. Therefore, the efficiency is evaluated as the sum of the coupled and the remaining power compared to the initial power:(5)$$\eta_{tot}=\frac{P_{probe}+P_{pump}}{P_{in}}$$

The related losses can then be estimated as:(6)$$a_{tot}=10\log_{10}\left(\eta_{tot}\right)\;\left[\mathrm{dB}\right].$$

This gives clues about the losses within the couling process and thus an indication of possible improvement.

The results in Figure 7 show an almost linear increase of the losses with applied force. Only for a crossing angle of $\alpha =90^{\circ }$, the losses are slightly larger and reach almost 8 dB for 2 N. This gives rise to the conclusion that the crossing angle can improve the capture of the light into the probe fibre, but the effect seems not really large.

### 3.4. Influence of Crossing Angle

The low coupling efficiencies can be mostly attributed to the weak acceptance of the leaking light from the pump fibre. Most of the light gets obviously out of the pump fibre, but is not collected by the probe fibre. For large angles, all light coming from the sides will just pass the fibre without being collected due to propagation angles larger than the critical angle within the fibre. In order to be captured, the propagation direction must be changed on its way through the probe fibre. This requires optical scattering or other mode-converting processes. Smaller angles between the two fibres will require less scattering and can potentially lead to higher capture efficiency. Therefore, the influence of crossing angle had been studied so that weaker scattering is needed for the required change of propagation. The experimental results for three applied forces of 0.86 N, 1.4 N and 1.96 N are shown in Figure 8.

## 4. Discussion

In this section, the measured results are compared to simulations, which can isolate individual effects. Therefore the contribution of these effects can be studied further and the relative importance of them can be assessed.

### 4.1. Influence of Deformation

The results show that the optical coupling between the fibres increase with applied force. This may come from the deformation and thus increased coupling area between the fibres. But the deformation can also lead to structural changes and a tapering of the fibre, which can cause additional scattering. In order to analyse the effect of the increased coupling area, the cross-coupled and the remain power, $P_{probe}$ and $P_{remain}$, were simulated for different deformations *d*. The coupling efficiency is plotted in Figure 9. It can be observed that the coupled power increases for higher deformations, in accordance to the measurement results. But for deformations larger than 40%, the efficiency declines again. The measurements, however, stayed in the almost linear region with much smaller deformations.

The simulated absolute values are also smaller with, for example, a crossing angle of $\alpha =30^{\circ }$ leading to about 0.3% coupling efficiency. This gives rise to the assumption that additional scattering or mode conversion must take place in the real experiment. This can be caused by the deformation, which decreases the total cross section of the fibre and mode conversion just in front of the coupling point. But the interface between the two fibres can also induce scattering.

Another aspect is the total remaining power $P_{pump}$ within the pump fibre. The experiments showed quite a strong loss, which reduced the remaining power to less than 40% even for moderate forces. The influence of the relative deformation on the remaining power is plotted in Figure 10. The same trend can be seen that the power is reduced for larger deformations. But there are also several differences: The amount of loss is considerably smaller with more than 90% of the original power even for deformations as large as 60%. This can be explained by the effect of deformation itself, which reduces the cross section of the pump fibre and has not been considered in the simulations. This effect seems to be dominant, in particular if one considers that the influence of the crossing angle is in the range of several percents. This influence is negligible compared to the drastic effect of the cross-section reduction, which was mainly observed in the experiments.

### 4.2. Influence of Crossing Angle

The crossing angle had a strong influence on the coupling efficiency. The more parallel the fibres were aligned to each other the larger the coupling was. A possible explanation is that the light, which coupled from the pump to the probe fibre, can be better captured and guided by the probe fibre if the crossing angle in smaller. Figure 11 illustrates the coupling between the fibres under different crossing angles. If the cross-coupled light enters the probe fibre under a too steep angle, which is more likely for large crossing angles α, it must be deflected at the coupling point so that the coupled light within the probe fibre can be guided again.

Therefore, simulations have been performed on the cross coupling at different crossing angles between $\alpha =15^{\circ }$ and $\alpha =75^{\circ }$. Figure 12 shows the obtained results. Here, the coupling efficiency is logarithmically plotted in the same way as the measurement results of Figure 8 in order to facilitate the comparison, but also because of the orders of magnitude difference between the results. Comparing both figures, it is obvious that the trend is the same. There are differences, though. The influence of the crossing angle seems to be greater. The coupling efficiency decreases to less than −60 dB for angles of $75^{\circ }$, whereas the measured efficiencies remained around −40 dB. This shows that there must be an additional effect, which improves the capturing of the coupled light within the probe fibre. The influence of the deformation seems to be slightly smaller, as well. This may also be attributed to the fact that the mode conversion at the transition between the straight fibre and the coupling section has not been considered. The change of the cross section will lead to a kind of tapering that increases the propagation angle, but also increases the power density at the coupling region. But this effect is not very strong compared to its influence on the remaining power in the pump fibre.

### 4.3. Influence of Interface Scattering

The investigations above show that the coupling efficiency cannot be explained by the deformation and the length of the coupling section alone. The cross-coupled light must be guided by the probe fibre in order to be detected as a sensor signal. Therefore, the propagation direction of the coupled ray must be within the limits of total internal reflection. Whereas this is often the case for small crossing angles and almost parallel fibres, the propagation angles become steeper and steeper for larger crossing angles. In order to be captured by the probe fibre, the coupled light must be deflected by scattering or other effects. Therefore, a certain scattering at the coupling interface between the fibres have been assumed. The actual propagation direction was changed by adding a random two-dimensional vector with a uniform random distribution along the circumference around the propagation direction and a normal distribution of the deviation Δθ with respect to the propagation direction. The influence of the scattering is shown in Figure 13 for crossing angles ranging between $\alpha =15^{\circ }$ and $\alpha =60^{\circ }$. For even larger angles, the simulated coupling efficiencies were too small provide any useful insight.

The simulations reveal that the influence of the scattering highly depends on the crossing angle, which makes sense. The larger the crossing angle is, the more deviation from the original propagation direction is needed in order to capture the most extreme rays. Thus for quite parallel fibres at an angle of $\alpha =15^{\circ }$, scattering rather reduces the coupling efficiency. There is a small local maximum at $\Delta \theta =10^{-2}$ rad, but more scattering reduces the efficiency even more. For larger crossing angles up to $30^{\circ }$, a similar trend can be observed, but the initial coupling efficiency is smaller due to the less parallel aligned fibres and the minimum occurs at larger scattering values. Then, the efficiency starts to increase again. For still larger crossing angles, the graph seems to be stretched even more towards larger scattering values. Here, scattering has a beneficial effect throughout and the efficiency only increases for larger scattering values. All this follows exactly the reasoning that for larger crossing angles more and more scattering is needed in order to actually capture the coupled light into the probe fibre. But there seems to be an optimum, which depends on the crossing angle and seems to be larger for higher crossing angles.

## 5. Conclusions

The coupling of light between two crossing fibres under load has been studied in detail for the application in a fibre-optical pressure sensor. Increased coupling for larger pressure could be verified. A study of the lost power in the pump and the collected power of the probe fibre revealed that most of the light is actually not captured. The simulations show that the deformation at the coupling point will lead to the majority of the losses. This effect has a two-fold influence: It decreases the optical power at the coupling point and thus reduces the coupling efficiency by that. But it also strongly attenuates the remaining power within the pump fibre, which will potentially be the input for a subsequent coupling point if a matrix of pressure sensors is realised. On the other hand, however, it can also lead to mode mixing, which may improve the coupling efficiency if properly designed.

The coupling efficiency could be drastically improved by adjusting the crossing angle between the fibres. A decease from $\alpha =90^{\circ }$ to $\alpha =15^{\circ }$ lead to 3% coupling efficiency, which is an improvement of more than 20 dB in sensitivity. This can be partially explained by the increased coupling area between the pump and probe fibre at smaller angles. However, the actual emission of light from the pump fibre follows a decaying exponential dependence with most light exiting at the beginning of the coupling section. The more effective influence comes from the fact that smaller crossing angles and thus more parallel fibres can capture the light more easily. The entering light from the pump fibre propagates at angles, which are further away from the critical angle of total reflection. Thus, further improvement can be achieved—in particular if the crossing angle has to be large—by scattering and mode mixing. But the optimum scattering depends on the crossing angle and may even deteriorate the coupling efficiency.

These results indicate that the pressure-dependent light coupling between fibres can also be applied for sensor arrays, which may be integrated into textiles. The observed large losses at the crossing points, which reduce the maximum number of sensor points, may be reduced by a proper design of the transition between the straight and the deformed fibre section. The coupling efficiency can be further improved by a controlled amount of scattering or mode conversion at the interface section. This may be realised by shaping the coupling region and a controlled scattering.

## Figures and Tables

**Figure 1 sensors-21-02402-f001:**
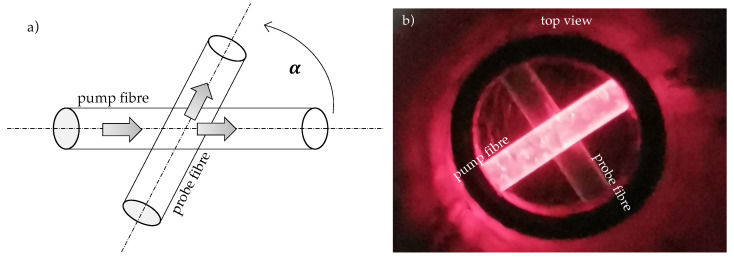
(**a**) Illustration of cross-coupling principle, (**b**) top-view photograph of cross-coupled fibres.

**Figure 2 sensors-21-02402-f002:**
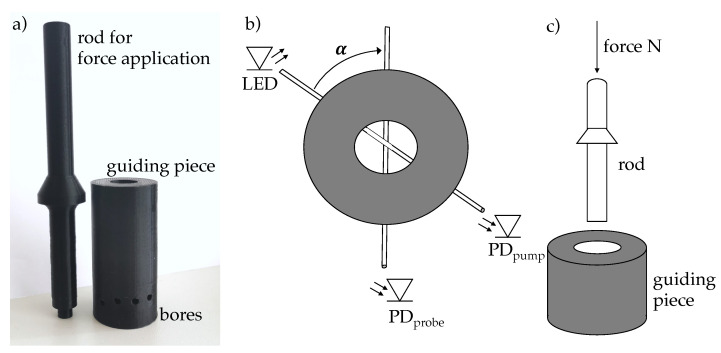
(**a**) 3D-printed coupling piece and force-application rod, (**b**) alignment of pump and probe fibre and (**c**) application of normal force to the fibre intersection.

**Figure 3 sensors-21-02402-f003:**
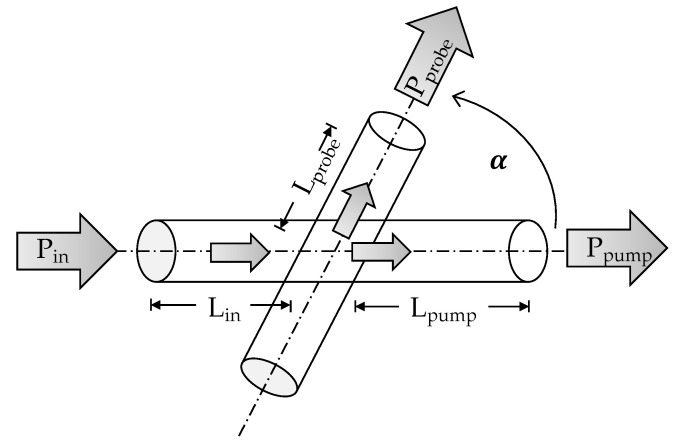
Schematic diagram of the actual measurement principle. An LED launched the power $P_{in}$ into the pump fibre. The fibre crossing was located after length $L_{in}$, where some of the power is coupled into the probe fibre and is received as $P_{probe}$ at the fibre end. The remaining pump power $P_{pump}$ was measured at the end of the pump fibre.

**Figure 4 sensors-21-02402-f004:**
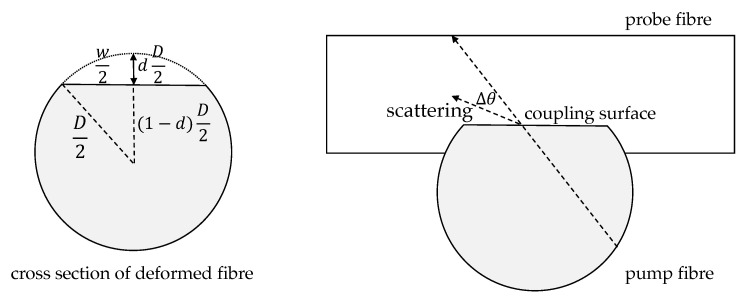
Schematic diagram of the deformed fibre cross section at the coupling point.

**Figure 5 sensors-21-02402-f005:**
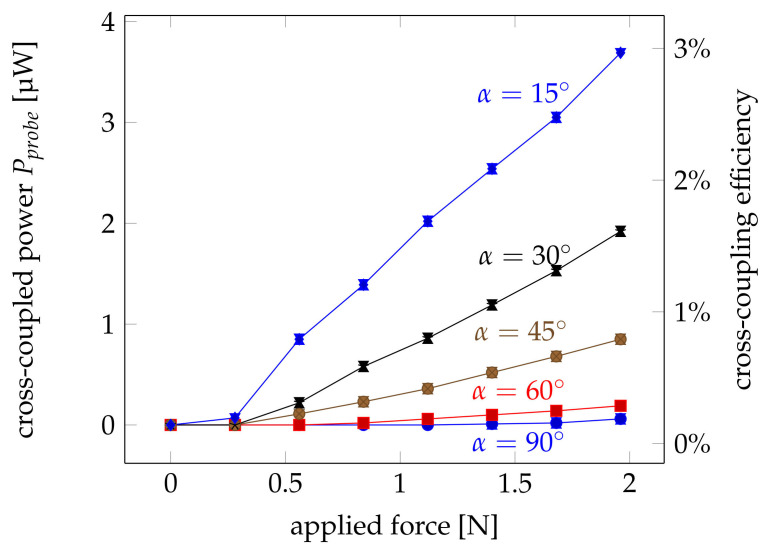
Measured coupled power $P_{probe}$ at the fibre end for different crossing angles between $\alpha =30^{\circ }$ and $\alpha =90^{\circ }$. The ordinate on the right-hand side represents the achieved cross-coupling efficiencies. Error bars indicate the 1σ deviation.

**Figure 6 sensors-21-02402-f006:**
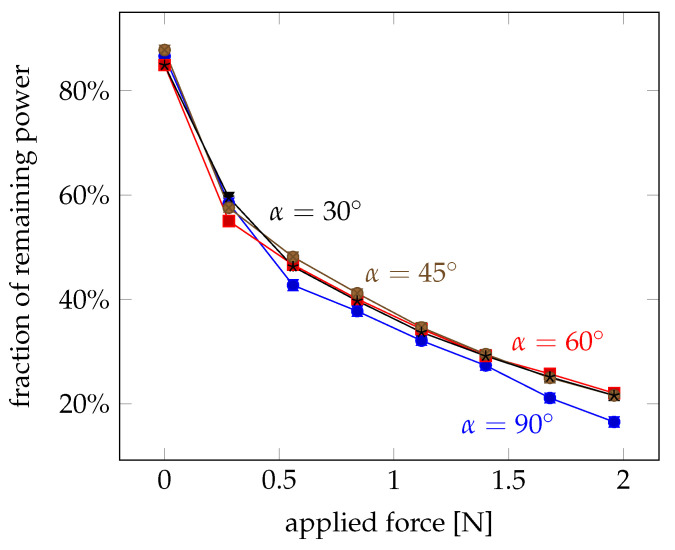
Remaining fraction of power in the pump fibre for different forces and crossing angles between $\alpha =30^{\circ }$ and $\alpha =90^{\circ }$. Error bars indicate the 1σ deviation.

**Figure 7 sensors-21-02402-f007:**
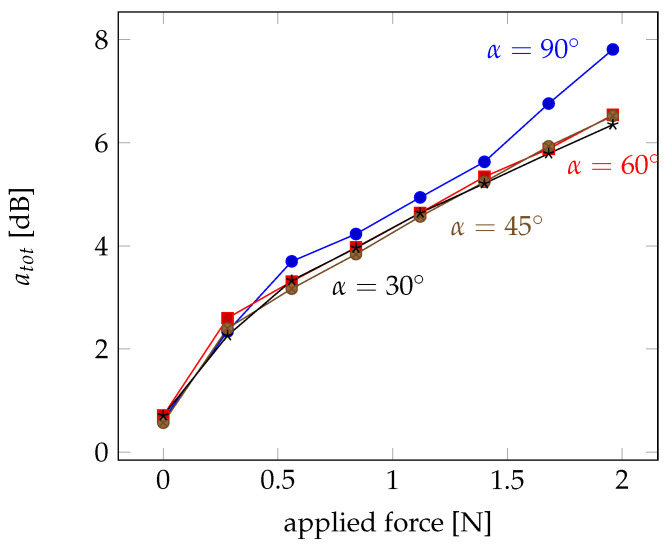
Losses at the coupling point for different forces and crossing angles between $\alpha =30^{\circ }$ and $\alpha =90^{\circ }$.

**Figure 8 sensors-21-02402-f008:**
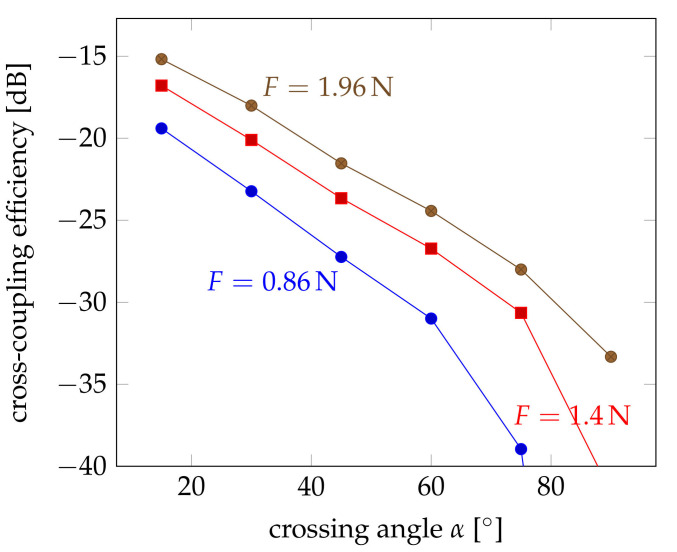
Influence of the coupling angle α on the cross-coupling efficiency at different loads of F∈{0.86N,1.4N,1.96N}.

**Figure 9 sensors-21-02402-f009:**
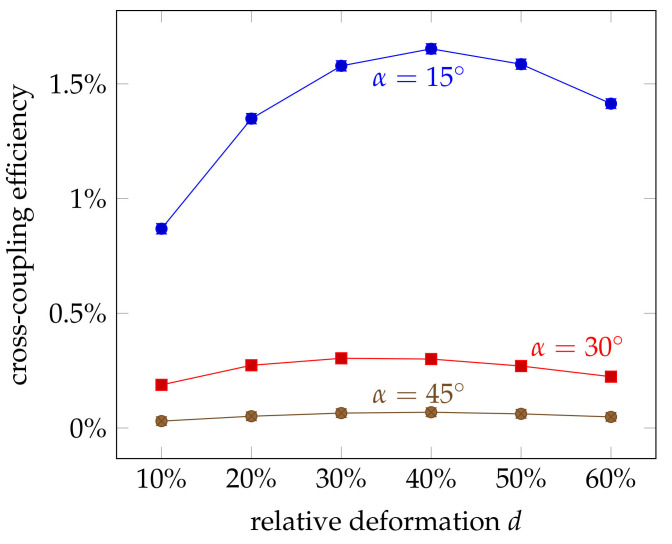
Influence of the relative deformation *d* on the cross-coupling efficiency at different crossing angles α and without scattering. For larger angles, the coupling efficiency showed the same trend but at a much lower level. Error bars indicate the 1σ deviation.

**Figure 10 sensors-21-02402-f010:**
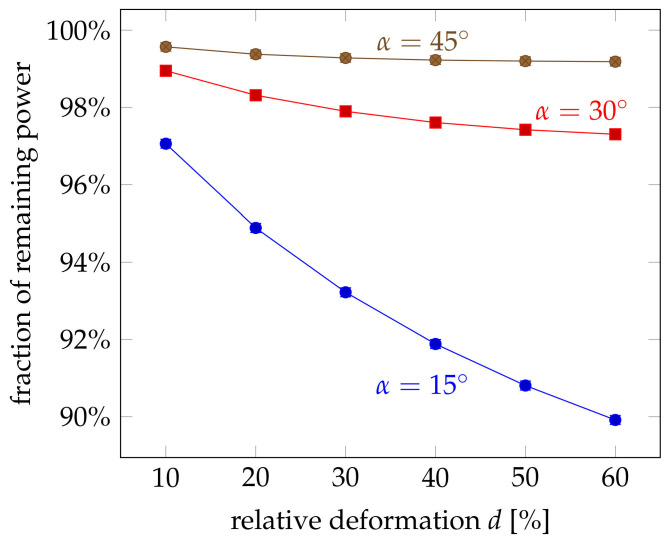
Influence of the relative deformation *d* on the cross-coupling efficiency at different crossing angles α and without scattering. For larger angles, the coupling efficiency showed the same trend but at a much lower level. Error bars indicate the 1σ deviation.

**Figure 11 sensors-21-02402-f011:**
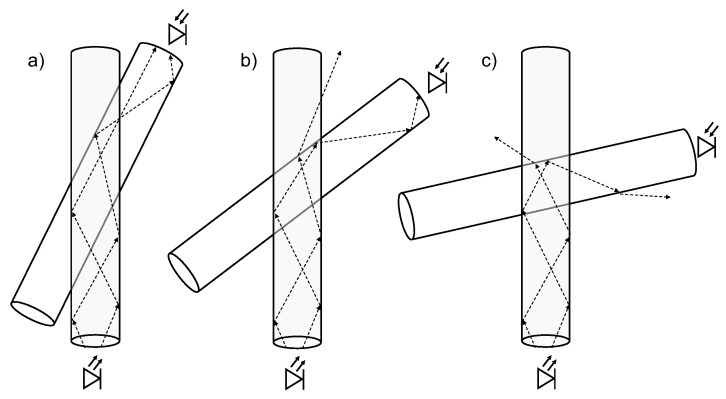
Exemplary ray paths in the cross-coupling between two optical fibres for three different coupling angles: (**a**) All rays are totally reflected and arrive at the photodiode. (**b**) Some of the rays are totally reflected; the rest is refracted. (**c**) All rays are refracted and do not reach the photodiode.

**Figure 12 sensors-21-02402-f012:**
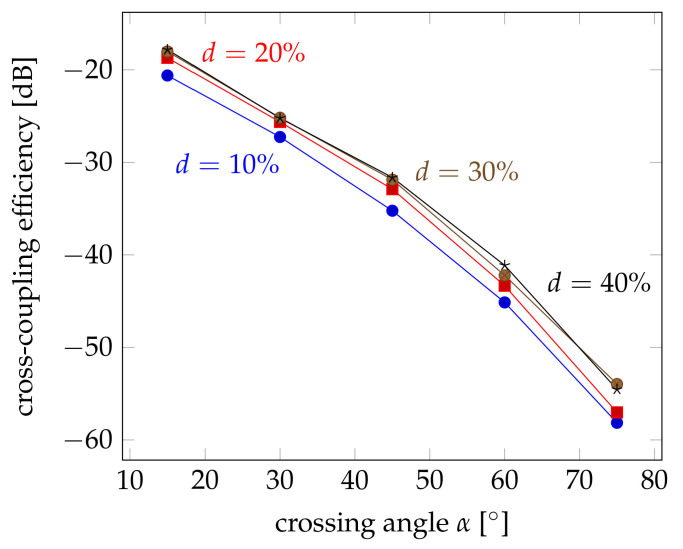
Influence of the crossing angle α on the cross-coupling efficiency for relative deformations up to d=40% and without scattering. For larger deformations, the coupling efficiency declined again.

**Figure 13 sensors-21-02402-f013:**
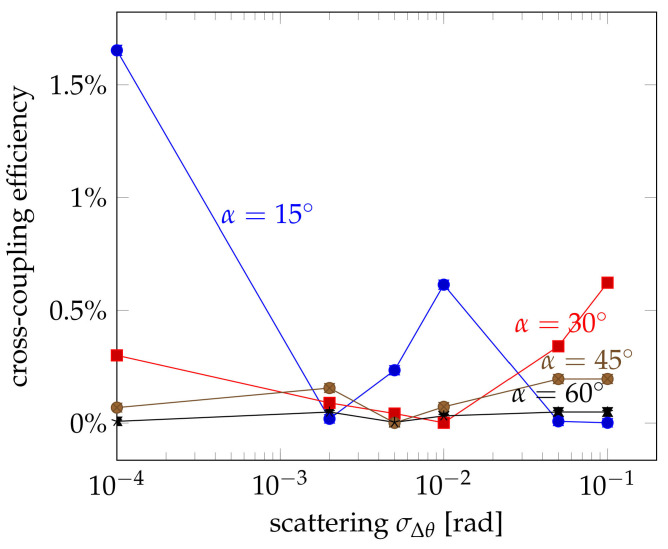
Influence of the interface scattering Δθ on the cross-coupling efficiency for different crossing angles α. All simulations were performed for a relative deformation of d=40%. Error bars indicate the 1σ deviation.

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
