# Peer review of "Analysis of Fibre Cross-Coupling Mechanisms in Fibre-Optical Force Sensorsâ€"

_sensors, 2021, doi:10.3390/s21072402_

Round 1
Reviewer 1 Report
Overall, I liked the manuscript. It is easy to read it, the material is presented clearly and consistently. The emphasis in the work is made on experimental research. At the same time, the technologies of calibration of such sensors and the method of multiplexing of arrays or a mesh of sensors are out of the study, It would be interesting for readers.
1. I have checked the references list. There are 17 references from the list are older than 10 years and 25 references from 32 in the list are older than 5 years. Despite the fact that the work looks fresh and interesting, it seems that the authors may have missed the analysis of the results of recent advances in the field of study. I recommend actualizing the review and including recent research results.
2. In the manuscript it is declared that the cross-coupling was investigated numerically. Despite this fact, the numerical model seems to be poor. The cross-section of deformed fiber does not correspond to the real physical deformation of the fibers. It looks like the very simplest model. In the real physical process, the deformation of the fiber will form a crosssection of the fiber-like to an ellipse. And coupling surface will form as two ellipses touching. It would be better to analyze the passing of light from pump fiber to the probe fiber taking into consideration the touching angles of two ellipses.
3. It is written in line 153 that the normal-distribution law was used for the light scattering, but the normal distribution is characterized by mean and dispersion. The estimation for dispersion value is not given.
4. The estimations of measuring errors are not given in the results section. Without the error estimations, the measurements can not be considered complete.
5. The conclusions seem evident. The smaller cross angle leads to the larger light transmission coefficient. The cross angle directly affects by the cross-sectional area of the fiber. The bigger cross-sectional area leads to a larger light transmission coefficient. The main result is in fact that 15° angle leads to more light transmission coefficient than bigger angles. I can suggest the following conclusion, for example, the angle in 5° leads to the bigger value of the light transmission coefficient. Is it correct also?
Author Response
Thank you very much for the comments and suggestions. We have written a detailed response in a separate document.

Reviewer 2 Report
This work presents an detailed analysis of optical fiber cross-coupling mechanisms in force sensors. I have some comments.
- In the introduction, from a detailed analysis, it is missing some recent papers about this matter in terms of pressure and force sensors, namely: a) b) Polymer optical fiber strain gauge for human-robot interaction forces assessment on an active knee orthosis, Optical Fiber Technology 41, 205-211,2019; b) Miniature all-fiber force sensor, Opt. Lett. 45, 5093-5096, 2020, c)Low-cost and high-resolution pressure sensors using highly stretchable polymer optical fibers, Materials Letters 271, 127810, 2020; c) Smart textiles for multimodal wearable sensing using highly stretchable multiplexed optical fiber system, Scientific Reports 10 (1), 1-12, 2020; d) A Miniature Triaxial Fiber Optic Force Sensor for Flexible Ureteroscopy, IEEE Transactions on Biomedical Engineering, 2021 (doi: 10.1109/TBME.2020.3034336). I think that the introduction must contain a nice state of the art for of pressure/force sensing.
2. The experimental results seem to show just in way (increasing) and not a cycle, like increasing and decreasing to observe the hysteresis of the measurements. Please, if posssible add more data for at leat 1 complete cycle (the better is to check 3 complete cycles to check the repeatability of the sensor.
3. The authors claim: "eformable polymer optical fibres made from polyurethane (PUR) were produced" how these fiber were produced? Nothing in good detail is mentioned.
4. What are the maximum number of sensors we can have and what is the influence in the performance? In additional, the humidity and temperature influence were taking in account and if not, what are the main concerns for the performance in a real application?
5. A clear understanding about simulation and experimental results must be well addressed.
Author Response
Thank you for the comments and suggestions. We compiled a separate document addressing the issues.

Round 2
Reviewer 1 Report
The main my comment was connected with the conclusions of the manuscript. The from 1 to 4 comments can be considered insignificant compared to 5, although they are important too. I have written in my last comment, that the conclusions seem evident. It is obvious, that the smaller cross angle leads to the larger light transmission coefficient. Moreover, It is evident, that the cross angle directly affects by the cross-sectional area of the fiber, and it is evident, that the bigger cross-sectional area leads to a larger light transmission coefficient.
The conclusions of the manuscript contain the statement, that "that the smaller cross angle leads to the larger light transmission coefficient" only. It is insufficient, in my humble opinion.
If you are sharing the conjecture that the bigger cross-sectional area leads to a larger light transmission coefficient, why not include this in the conclusions?
We will assume that I am satisfied with the answers to comments 1-4. Although I have doubts about the repeatability of the results within 1%, I will not dispute this. At the same time, the microscopic negligible changes of the manuscript with one well-known formula of Normal distribution from Wikipedia adding are not sufficient to make the manuscript better.
You see, colleagues, the current look of the manuscript makes for a frivolous look for your serious study. One gets the impression that this is student laboratory work. It seems to me that the conclusions of the manuscript should be seriously revised.
Author Response
Dear reviewer,
thank you for the comments. We prepared a document that address all your concerns.
Best regards,
Christian Bunge

Reviewer 2 Report
The paper was very well improved and the suggestions and doubts were well addressed. Well done.
Author Response
Dear reviewer,
we are pleased that the revisions according to your suggestions have been well received.
Best regards,
Christian Bunge